# Strong vibrational coupling in room temperature plasmonic resonators

Junzhong Wang[1], Kuai Yu [1], Yang Yang[1], Gregory V. Hartland[2], John E. Sader[3] & Guo Ping Wang[1]

Strong vibrational coupling has been realized in a variety of mechanical systems. However, there have been no experimental observations of strong coupling of the acoustic modes of plasmonic nanostructures, due to rapid energy dissipation in these systems. Here we realized strong vibrational coupling in ultra-high frequency plasmonic nanoresonators by increasing the vibrational quality factors by an order of magnitude. We achieved the highest frequency quality factor products of $f \times Q = 1.0 \times 10^{13}$ Hz for the fundamental mechanical modes, which exceeds the value of $0.6 \times 10^{13}$ Hz required for ground state cooling. Avoided crossing was observed between vibrational modes of two plasmonic nanoresonators with a coupling rate of $g = 7.5 \pm 1.2$ GHz, an order of magnitude larger than the dissipation rates. The intermodal strong coupling was consistent with theoretical calculations using a coupled oscillator model. Our results enabled a platform for future observation and control of the quantum behavior of phonon modes in metallic nanoparticles.

[1] College of Electronic Science and Technology, Shenzhen University, Shenzhen 518060, China. [2] Department of Chemistry and Biochemistry, University of Notre Dame, Notre Dame, IN 46556, USA. [3] ARC Centre of Excellence in Exciton Science, School of Mathematics and Statistics, The University of Melbourne, Melbourne, VIC 3010, Australia. Correspondence and requests for materials should be addressed to K.Y. (email: kyu@szu.edu.cn) or to G.P.W. (email: gpwang@szu.edu.cn)

 **1**

The observation of quantum effects in mechanical systems requires high-quality factor resonators that can be cooled to their ground state[1]. The temperatures needed to achieve this are $T < hf/k_B$, where $f$ is the vibrational frequency, $h$ and $k_B$ are Planck's and Boltzmann's constants, respectively[2–4]. The majority of the systems that have been studied to date have been nanofabricated dielectric or semiconducting devices, with frequencies in the kHz to few GHz range[5–11]. Experimentally cooling such low-frequency mechanical resonators to their quantum ground state is an enormous challenge, requiring cryogenic temperatures and cooling via radiation pressure. However, nanomaterials support vibrations at ultra-high frequencies (>50 GHz) and, thus, may enable the observation of the quantum regime for mechanical oscillators at moderate temperatures. A benchmark for evaluating whether a mechanical system can be cooled to its ground state is the frequency quality factor product $f \times Q$. This product should be greater than $k_B T/h$, that is, the mechanical quality factor $Q$ must be larger than the number of thermal phonons at the ambient temperature ($Q > k_B T_{room}/fh$)[12–15].

A major issue for resonators based on nanoparticles is actuating the vibrations and reading out the response. For metallic resonators, actuation can be achieved by exciting the plasmon resonances of the nano-object. Decay of the plasmon oscillation causes rapid heating that impulsively excites vibrational modes of the particles[16]. However, these plasmonic nanoresonators suffer from both intrinsic and environmental energy dissipation mechanisms that reduce the vibrational quality factors. The intrinsic damping effect can be reduced by using single crystal nanoparticles created by chemical synthesis, rather than the polycrystalline particles produced by lithography[17,18]. Environmental damping for plasmonic nanoresonators predominantly occurs by radiation of acoustic waves into the surroundings[19]. Blocking the out-propagating acoustic waves and confining the energy to the resonators will be a major step to creating high vibrational quality factors for these systems.

Constructing high-frequency/high-quality factor plasmonic nanoresonators will be attractive for cavity optomechanics and electromechanics applications[4,20], where strongly coupled systems with low losses are needed to observe effects such as Rabi splitting and electromagnetically induced transparency[21,22]. However, the large damping rates that have been reported for plasmonic resonators to date makes strong coupling an unattainable regime[23–26]. Here we improved the vibrational quality factor of Au nanoplates by an order of magnitude by blocking the out-propagating acoustic waves. The resonators have mechanical fundamental modes with average frequency quality factor products of $f \times Q = 1.0 \times 10^{13}$ Hz at room temperature. Strong coupling between the vibrational modes of two nanoplates was observed with a coupling rate $g = 7.5 \pm 1.2$ GHz and a value of $g/\omega = 0.14$ was obtained indicating the coupling strength is comparable to the natural frequency of the mechanical resonator. The observation of strong vibrational coupling between two plasmonic nanoresonators has not been previously reported, and is an important step for achieving quantum control of the mechanical modes of nanostructures.

## Results

### Quality factor improvement of plasmonic nanoresonators.
Au nanoplates were chemically synthesized based on previous studies (see Methods for details)[27]. The majority of the sample was made up of hexagonal and triangle plates with average edge lengths of 10–20 µm as shown in Supplementary Fig. 1. The thickness of the Au nanoplates was determined by atomic force microscopy (AFM), and a representative AFM image is shown in

Supplementary Fig. 2. Note that the in-plane shape had no influence on the thickness-dependent mechanical vibrations and damping for the large aspect ratio nanoplates in this work. The crystallographic structure of the Au nanoplates was characterized and gave a hexagonal symmetry diffraction pattern demonstrating single crystal nanoplates where the surfaces are {111} planes, as shown in Supplementary Fig. 3[27,28].

Mechanical vibrations of Au nanoplates were launched by 800 nm femtosecond pulsed lasers and monitored with a 530 nm probe beam in a pump–probe scheme (see Methods for details)[29]. The Au nanoplates were deposited on either glass substrates or Lacey carbon films as schematically illustrated in Fig. 1a, d, respectively. Figure 1b shows a transient absorption trace for a Au nanoplate on the glass substrate where pronounced modulations are observed superimposed on an exponentially decaying background. The modulated signal is assigned to Brillouin oscillations that arise from the interaction of light with propagating picosecond acoustic waves in the glass[30]. The formation of out-propagating picosecond acoustic waves demonstrates that the substrate is strongly mechanically coupled to the nanoplate. In the current studies, the experimental traces were fitted to the function:

$$\Delta I(t) = \sum_{k=(el,ph)} A_k \exp\left(-\frac{t}{\tau_k}\right) + \sum_{n=(1,2,\ldots)} A_n \cos\left(\frac{2\pi t}{T_n} - \phi_n\right) \exp\left(-\frac{t}{\tau_n}\right),$$
(1)

where the first term accounts for the background signal due to cooling of the nanoplate from electron-phonon ($k = el$) and phonon-phonon ($k = ph$) interactions, and the second term accounts for various vibrations with $n = 1, 2, \cdots$ representing the number of modes. Specifically, Fig. 1b was fitted to Eq. (1) with one oscillation term with a period $T_b = 30.83 \pm 0.02$ ps and damping time $\tau_b = 556 \pm 32$ ps. This signal is assigned to Brillouin oscillations in glass. A Fast Fourier transform (FFT) of the data is shown in Fig. 1c. The frequency $f_b = 32.44 \pm 0.02$ GHz and damping constant $\Gamma = 1.8 \pm 0.1$ are consistent with the time domain results.

The frequency of the Brillouin oscillations depends on the refractive index and speed of sound of the material[31–33]. Specifically, the Brillouin oscillation frequency ($f_b$) and the wavelength of the acoustic waves ($\lambda_b$) are:

$$f_b = 2v_l n \cos\phi/\lambda_{pr},$$
(2a)

$$\lambda_b = v_l/f_b = \lambda_{pr}/2n\cos\phi,$$
(2b)

where $v_l$ is the longitudinal sound velocity in the medium, $n$ is the refractive index of the medium, $\phi$ is the angle of incidence of the probe beam, and $\lambda_{pr}$ is the wavelengths of the probe beam. Using a refractive index $n = 1.46$ of glass and Brillouin oscillation frequency $f_b = 32.44$ GHz at 530 nm, we calculated a longitudinal speed of sound $v_l = 5900$ ms$^{-1}$ and acoustic wavelength $\lambda_b = 180$ nm for normal incidence, which is consistent with previous measurements[30]. The coefficient of acoustic wave attenuation is $\alpha = \Gamma\pi/v_l = 0.95 \pm 0.05$ µm$^{-1}$ which is larger than the literature value for glass due to diffraction effects[30,34].

The transient absorption trace in Fig. 1b only shows Brillouin oscillations—the localized acoustic vibrations are completely absent. In general, only a fraction of the Au nanoplates on the glass substrate (<30%) display localized acoustic vibrations. This is attributed to strong damping of the acoustic modes by the glass substrate. The occasional appearance of the acoustic modes could be due to the presence of surfactant, which insulates the nanoplates from the glass substrate[19]. Supplementary Fig. 4 shows FFT spectra where both Brillouin oscillations ($f_b = 32.1 \pm 0.7$ GHz for all Au nanoplates) and a higher

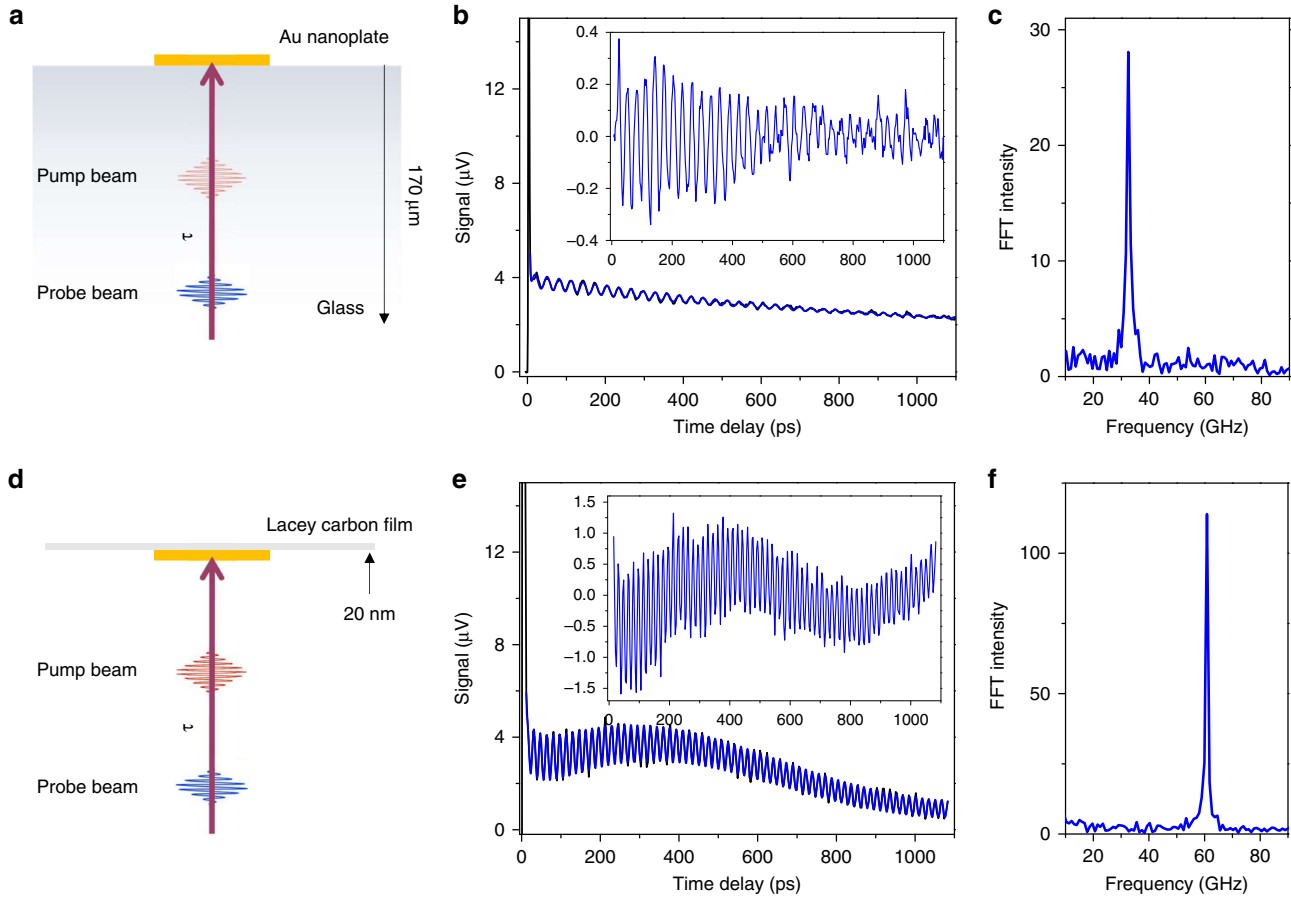

**Fig. 1** Brillouin oscillations and localized acoustic vibrations of Au nanoplates. **a** Diagram of experimental geometry for Brillouin oscillation detection where the Au nanoplates were deposited on glass substrate. **b** Transient absorption trace of a Au nanoplate where Brillouin oscillations are observed. The blue line is the fitting curve to the experimental data with one oscillation term, see Eq. (1). The inset shows the isolated Brillouin oscillation component. **c** Fast Fourier transform (FFT) of the Brillouin oscillations. **d** Diagram of experimental geometry for localized acoustic vibration detection where the Au nanoplates were deposited on Lacey carbon film. **e** Transient absorption trace of acoustic vibrations. The blue line is the fitting curve to the experimental data with two oscillation terms. The inset shows the isolated acoustic vibrations. **f** FFT of the acoustic vibrations. The glass substrate and Lacey carbon film have thickness of 170 μm and 20 nm, respectively

frequency peak that is assigned to the breathing modes can be observed. The measured frequencies for the breathing modes vary from plate to plate due to differences in thickness, and are severely broadened with an average quality factors $Q_{br} = 10 \pm 3$ (see Fig. 2 below). The low-quality factor for this sample is consistent with previous studies of nanoparticles on a glass surface[28,30,35]. Energy redistribution from the localized acoustic vibrations into the propagating sound waves that give rise to the Brillouin oscillations results in severe damping of the mechanical modes. Thus, an improvement in the vibrational quality factors could be achieved if this energy flow pathway could be blocked.

Acoustic impedance mismatch is the major factor for controlling the flow of acoustic energy[19,36]. This implies that using porous low-density materials for the substrate could be an effective way to increase the vibrational quality factors of metallic nanoresonators. Thus, Lacey carbon films were used to replace the glass substrates (Fig. 1d). The porous (~5 μm pore size) and thin (20 nm) carbon film provided a robust support of Au nanoplates as shown in Supplementary Fig. 5. Previously, trenches were used to isolate metal nanostructures from the substrate to improve the vibrational quality factors[37–39]. This design produced moderate quality factors of 40–60 for the breathing modes of Au nanowires, and ~30 for Au nanoplates with thicknesses of several hundred nanometers[28]. Figure 1e shows a transient absorption trace for a Au nanoplate on a

Lacey carbon film where pronounced modulations from the breathing mode associated with changes in the width of the nanoplate can be observed. The experimental trace was fitted to Eq. (1) with two damped harmonic oscillations. The high-frequency oscillation was assigned to the breathing mode and the other low-frequency oscillation to a "bouncing" mode (motion of the nanoplate relative to the substrate)[40,41]. The fit yields oscillation periods $T_{br} = 16.45 \pm 0.003$ ps, $T_{bo} = 1650 \pm 60$ ps and damping times $\tau_{br} = 1028 \pm 75$ ps, $\tau_{bo} = 1080 \pm 300$ ps for the breathing mode and bouncing mode, respectively. This gives a quality factor for the breathing mode of $Q_{br} = \pi\tau_{br}/T_{br} = 196 \pm 15$. For the bouncing mode the quality factor was on the order of 1; however, the error is large due to the limited scanning range of the delay line in our experiments. We therefore focus on the breathing mode vibrations. Figure 1f shows the Fourier transform of the data in Fig. 1e, which yields a breathing mode vibration frequency $f_{br} = 60.76$ GHz with damping constant $\Gamma = 0.85$ GHz. This analysis yields a quality factor of $Q_{br} = 227 \pm 11$ in reasonable agreement with value derived from fitting the transient absorption trace[42]. In the following analysis the quality factors were obtained from fitting the transient absorption traces with Eq. (1). Note that the measured quality factor for the nanoplate in Fig. 1e is the highest value reported so far for plasmonic resonators at ambient conditions[37,38].

The measured vibrational frequency is independent of the in-plane shape for such large aspect ratio nanoplates and expected to have thickness dependence of $f_{br} = v_l/2d$, where $v_l = 3240$ ms$^{-1}$ is the longitudinal speed of sound of bulk gold[28,35]. We estimate a thickness of $d = 27$ nm for a vibrational frequency $f_{br} = 60.76$ GHz. Figure 2 shows FFT spectra and vibrational quality factors of Au nanoplates in a broad frequency range supported on Lacey carbon films. The FFT spectra exhibited localized acoustic vibrations with narrow bandwidth for all of the measured nanoplates. The vibrational frequencies vary from 30 to 80 GHz (average = $55 \pm 10$ GHz) corresponding to plate thicknesses of 20–50 nm. These results are consistent with AFM statistical measurements of the sample.

The exceptionally narrow vibrational bands in Fig. 2 have an average quality factor $Q_{br} = 180 \pm 26$. Compared to Au nanoplates on glass substrate with $Q_{br} = 10 \pm 3$, there is an order of magnitude increase in vibrational quality factor. Importantly, the Au nanoplates exhibit average frequency quality factor products of $f \times Q = 1 \times 10^{13}$ Hz, which is larger than the value of $k_B T_{room}/h = 0.6 \times 10^{13}$ Hz required for ground state cooling at room temperature[12–14]. This implies that the mechanical quality factors surpass the number of room temperature thermal phonons, $Q > \tilde{n}_t = k_B T_{room}/hf$[12,13], which is the benchmark for

observing quantum effects in mechanical systems. We also note that the signal-to-noise ratio (SNR) for the FFT spectra of the nanoplates on the Lacey carbon film is $60 \pm 10$, which is significantly larger than the value of $15 \pm 5$ for the glass substrate (see Supplementary Fig. 4). In applications where nanoelectromechanical or nanooptomechanical systems are used for force and/or mass detection, the frequency noise in the measurement is given by $\Delta f/f \sim \frac{1}{2Q}\frac{1}{SNR}$[43]. The large quality factors and high SNR for the present materials mean that they are promising candidates for sensing applications[44,45], exceeding the performance of traditional plasmonic nanoresonators by several orders of magnitude[46,47].

**Strong vibrational coupling.** The narrow linewidths for the Lacey carbon film supported nanoplates means that they are ideal systems to study coupling between mechanical resonators. Examples of vibrational coupling between overlapping Au nanoplates are shown in Fig. 3 and Supplementary Fig. 6. Figure 3a shows transient absorption traces recorded for a pair of Au nanoplates in the overlap regime, and in regions where the plates do not overlap. The corresponding FFT spectra are shown in Fig. 3b. The measurements in the non-overlapping region show that the two Au nanoplates have fundamental vibrational

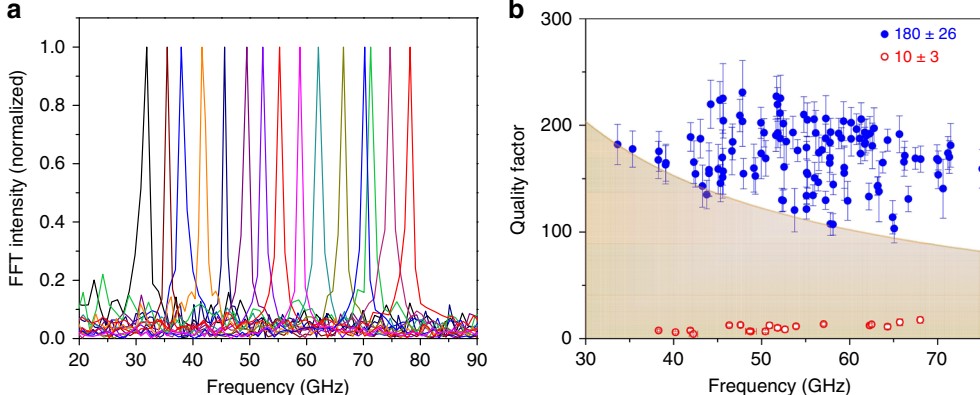

**Fig. 2** Fast Fourier transform (FFT) spectra and quality factors $Q_{br}$ of Au nanoplate vibrations on Lacey carbon films. **a** Thickness-dependent breathing mode vibrations. **b** Quality factors $Q_{br}$ for the different nanoplates. The average quality factor is $180 \pm 26$, where the error is the standard deviation. For comparison, $Q_{br}$ is $10 \pm 3$ for Au nanoplates on glass substrates. The solid line on top of the shaded area corresponds to mechanical vibration quality factor $Q = k_B T_{room}/hf_{br}$. Note that the quality factors were retrieved by fitting to Eq. (1), not from the FFT analysis

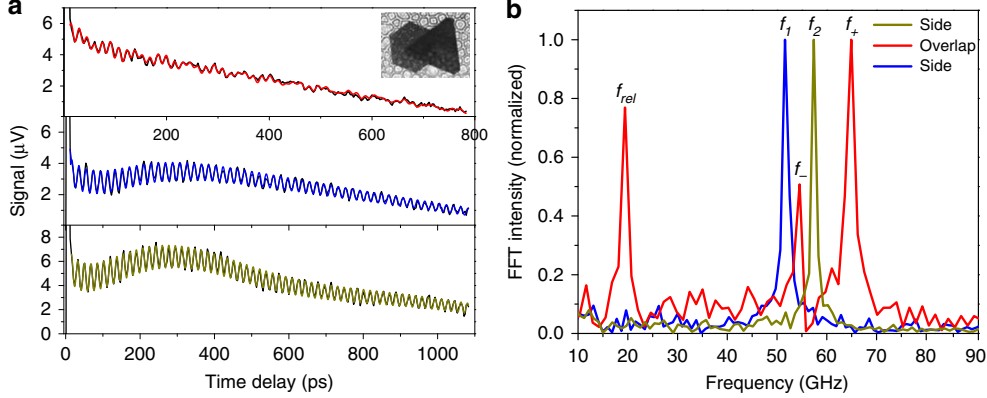

**Fig. 3** Strong vibrational coupling for stacked Au nanoplates. **a** Transient absorption traces for stacked Au nanoplates probing on each single nanoplate and the overlapping area. The color lines are the fitting curves to the experimental data. The inset shows an optical image of the stacked Au nanoplates on carbon film. **b** Fast Fourier transform (FFT) spectra of the mechanical vibrations of the first plate $f_1$, second plate $f_2$, and the overlapping area. Mechanical coupling between the plates creates new frequencies $f_+$ and $f_-$. The vibration at ~20 GHz was ascribed to a mode that corresponds to motion of the two nanoplates relative to each other

frequencies $f_1 = 51.35$ GHz and $f_2 = 57.12$ GHz, respectively, with damping rates $\Gamma_1, \Gamma_2 \approx 1$ GHz. Mechanical coupling is clearly observed in the FFT spectrum for the overlapping region as a shift in the vibrational frequencies to a higher mode $f_+ = 64.60$ GHz and a lower mode $f_- = 54.26$ GHz. The higher mode has a frequency increase of $f_+ - f_2 = 7.48$ GHz, which exceeds the damping rates $\Gamma_1, \Gamma_2$, indicating a strong coupling in these plasmonic nanoresonators. The complete data for all the coupled resonators investigated in this study are presented in Supplementary Table 1. We determined the system errors by measuring the same Au nanoplate multiple times at different positions as shown in Supplementary Fig. 7. The standard deviation of measured frequencies for an isolated Au nanoplate was $52.69 \pm 0.12$ GHz (0.2%), while it was $74.58 \pm 0.84$ GHz (1.1%) for the coupled nanoplates. The large spread of measured coupling frequencies could be due to the inhomogeneous environments, such as differences in the amount of PVP between Au nanoplates which could affect the coupling strength. The phase difference between the coupled modes $f_+$ and $f_-$ is presented in Supplementary Fig. 8. There is a negligible phase difference, which indicates the two modes are normal modes of the system that are excited by the same excitation mechanism (ultrafast pump laser-induced heating).

The experimental results for the coupled resonators were modeled using the classic damped harmonic oscillator model. Each Au nanoplate $n$ (with $n = 1, 2$) was assigned an effective mass $m_n$, stiffness $k_n$ and dissipation rate $\Gamma_n$. The coupling element consists of a spring constant $k_c$ and a damping rate $\Gamma_c$, as shown in the schematic of the coupled resonators presented in Fig. 4. The transition from weak to strong coupling depends on the spring constant $k_c$ for coupling. To determine the spectrum, both the Au nanoplates in the model were subjected to a time-dependent external force $F(\omega) = Fe^{-i\omega t}$. The dynamics can be expressed by the following differential equations in terms of the displacements of $x_1$ and $x_2$ of the oscillators from their respective equilibrium positions:[8,48,49]

$$\ddot{x}_1 + \gamma_1 \dot{x}_1 + \omega_1^2 x_1 + v_{12}(x_1 - x_2) + \gamma_{12}(\dot{x}_1 - \dot{x}_2) = Fe^{-i\omega t}, \quad (3a)$$

$$\ddot{x}_2 + \gamma_2 \dot{x}_2 + \omega_2^2 x_2 + v_{21}(x_2 - x_1) + \gamma_{21}(\dot{x}_2 - \dot{x}_1) = Fe^{-i\omega t}, \quad (3b)$$

where $\omega_n = \sqrt{k_n/m_n}$ are the mode frequencies, $\gamma_n = \Gamma_n/m_n$ are the energy dissipation rates, and $v_{12} = v_{21} = \sqrt{k_c/m_n}$ and $\gamma_{12} = \gamma_{21} = \Gamma_c/m_n$ are the intermodal coupling and damping coefficients, respectively. The solutions of the displacement $x_n(t)$ are assumed in the form of $x_n(t) = X_n(\omega)e^{-i\omega t}$. The nontrivial solution of the equations yields eigenfrequencies[49]

$$\omega_{\pm}^2 = \frac{1}{2}\left[\omega_1^2 + \omega_2^2 + 2v_{12} \pm \sqrt{(\omega_1^2 - \omega_2^2)^2 + 4g^2\sqrt{(\omega_1^2 + v_{12})(\omega_2^2 + v_{12})}}\right],$$

where the coupling strength $g = v_{12}/\sqrt[4]{(\omega_1^2 + v_{12})(\omega_2^2 + v_{12})}$, $\omega_{\pm}$ are the oscillator frequencies for the two Au nanoplates in the presence of mutual coupling.

The calculated vibrational spectra are shown in Fig. 4a for mechanical resonators with different coupling rates. The vibrational spectra of the individual resonators are shown as the dotted lines; these spectra overlap the calculated spectra from Eq. (3) for $g = 0$ (no coupling). Importantly, the spectra are dramatically shifted when coupling is introduced into the simulations. The measurements in Fig. 3b can be qualitatively reproduced by simulations with coupling rate $g = 8$ GHz.

A statistical analysis of the experimental data is presented in Fig. 4b and Supplementary Fig. 9a where the frequency shifts $f_+ - f_2$, $f_- - f_1$ and coupling strength $g$ are plotted versus the fundamental frequency detuning $\Delta_{12} = f_2 - f_1$. An average intermodal coupling rate of $g = 7.5 \pm 1.2$ GHz was obtained from the experimental measurements. A plot of $f_+$ and $f_-$ versus $\Delta_{12}$ is presented in Supplementary Fig. 9b for coupled resonators with $f_1 \approx 60$ GHz. The data show an avoided crossing with a Rabi splitting frequency of ~7.5 GHz, consistent with theoretical calculations and analysis in Fig. 4. Note that the coupling rate exceeds the dissipation rates of the uncoupled oscillators by an order of magnitude, showing that the system is well within the strong coupling limit. The strength of the intermodal coupling can also be quantified by the cooperativity, which is defined as $C = 4g^2/\Gamma_1\Gamma_2$[1,10]. The data show a value of $C = 225$, which again indicates strong coupling for this system.

The vibrational quality factors for the coupled vibrational mode are shown in Supplementary Fig. 10a. The average value is $Q = 95 \pm 30$. The increased attenuation for the coupled system is probably because the overlapped nanoplates introduce additional relaxation channels compared to the isolated nanoplates. Besides the mechanical coupling for the overlapped Au nanoplates, there is a vibrational mode $f_{rel} = 19.38$ GHz in Fig. 3b which was ascribed to a mode arising from relative motion of the two nanoplates[50]. This mode appeared for all of the coupled Au nanoplates, as is listed in Supplementary Table 1. The relative motion mode has a vibrational frequency in the 10–20 GHz range, and a quality factor of $Q_{rel} = 21 \pm 7$ (see Supplementary Fig. 10). Analysis of the relative motion mode gives the characteristic cut-off frequencies of $f_a = 24.4 \pm 1.2$ GHz which

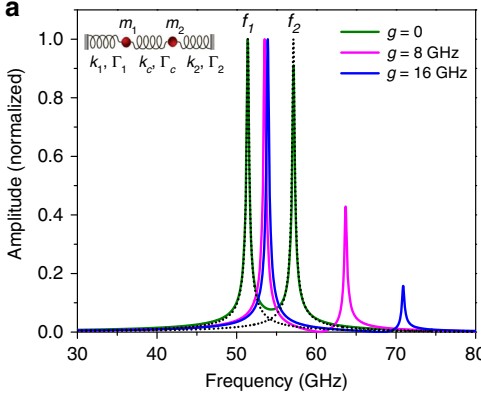

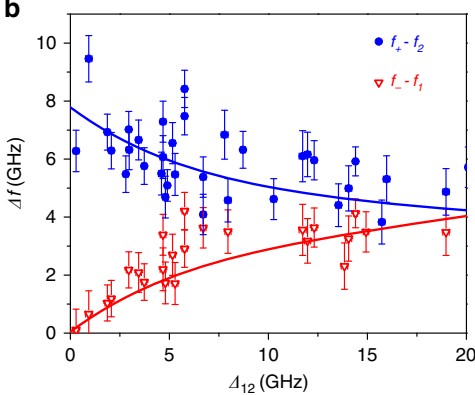

**Fig. 4** Simulations of mechanical coupling between resonators. **a** Calculated vibrational spectra with different coupling rates. The inset shows the schematic model of coupled resonators. **b** Frequency shift of the higher $f_+$ and lower $f_-$ modes versus frequency detuning $\Delta_{12} = f_2 - f_1$. The symbols are experimental results with standard deviations of 1.1% and the lines are calculated frequency shifts for the coupling rate $g = 7.5$ GHz

corresponds to bond spring constant $\alpha \approx 8 \times 10^{18}$ Nm$^{-3}$ between Au nanoplates[50].

The vibrational coupling is insensitive to the excitation power as shown in Supplementary Fig. 11, where data from coupled nanoplates recorded with different intensity pump pulses are presented. The vibrational amplitude increases with increasing the pump power; however, the FFT spectra have identical vibrational frequencies. Note that the vibrational coupling is highly sensitive to the environment. Mechanical coupling between Au nanoplates was not observed when the nanoplates were immersed in water, as shown in Supplementary Fig. 12. The lower frequencies at ~7.4 GHz observed for the nanoplates in water corresponds to the Brillouin oscillations in water. The value of $f_b$ = 7.4 GHz for water yields a longitudinal speed of sound of $v_l$ = 1470 ms$^{-1}$ assuming $n$ = 1.33, which is consistent with our previous measurements[30]. The intermodal coupling can be partially restored after evaporating the water, as shown in Supplementary Fig. 13.

## Discussion

Creating high-quality factor mechanical resonators in the ultra-high-frequency range (GHz–THz) is interesting for many applications, ranging from mass sensing to quantum mechanics[1,45]. Even though plasmonic nanoresonators can achieve high vibrational frequencies, they suffer from both intrinsic and environmental dissipation effects and, thus, typically have small quality factors[51]. In general, the total quality factor for a given vibrational mode can be expressed as $\frac{1}{Q_{total}} = \frac{1}{Q_{int}} + \frac{1}{Q_{env}}$, where $Q_{int}$ is the intrinsic damping quality factor, and $Q_{env}$ corresponds to the environment damping. Normally intrinsic damping of chemically synthesized nanoparticles is relatively small[17,39], although it may become dominant with lithographically fabricated plasmonic nanostructures where the crystal defects were abundant[18]. The measured quality factor $Q_{total}$ = 180 ± 26 for Au nanoplates on Lacey carbon films is remarkable, especially considering that molecular capping ligands were presented, as can be seen from the TEM measurements in Supplementary Fig. 3. Previous experiments on Au nanowires have measured quality factors for damping by the surfactant layer of $Q_{surf} \approx 200$[19,39], which implies that the intrinsic damping for the chemically synthesized thin Au nanoplates in this study must be very small[17]. This also means that it may be possible to further improve the quality factor by fully removing the surface-capping layer.

Environmental damping is very dependent on the energy transfer efficiency between the localized acoustic vibration modes and sound waves in the surrounding medium[19,36]. The quality factor $Q_{total}$ = 10 ± 3 for Au nanoplates on glass substrates indicates severe damping from the generation of acoustic waves in the glass, which can be detected as Brillouin oscillations. Replacing the glass substrate with 20 nm Lacey carbon films greatly improved the vibrational quality factors by blocking energy transfer to propagating longitudinal acoustic waves. Previously, suspending metal nanowires over trenches was used to improve the quality factors of plasmonic resonators[28,38,39]. However, the quality factors were much smaller than these measured here, which could be due to the effects from the contact points for the nanowires or from difference in $Q_{int}$[37,38]. High-quality factors were also recently reported for gold disks created by nanolithography, and attributed to a hybridization effect that created vibrational modes that are effectively decoupled from the substrate[42]. However, the frequency quality factor products for these nano-objects remained ~$0.1 \times 10^{13}$ Hz. Supporting the plasmonic structures with Lacey carbon films thus is an efficient method for blocking the acoustic waves and increasing the quality factors. Indeed, the quality factors of >200 and frequency quality factor products of >$1.0 \times 10^{13}$ Hz observed in this study are the highest

that have been reported to date for plasmonic nanoresonators. These results are an important step for achieving phonon engineering. Note that the propagation distance of acoustic waves in the lateral dimensions of Au nanoplates is ~3 μm for the 1 ns vibrational lifetime and speed of sound $v_l$ = 3240 ms$^{-1}$ in gold. This propagation distance is only slightly larger than the excitation spot (~1 μm), which indicates that flow of acoustic energy out of the excitation region should not be an issue in these experiments.

The improvement in the quality factors and SNR is beneficial for realizing strong coupling in plasmonic resonators. Specifically, we were able to observe coupling between overlapped Au nanoplates (average vibrational frequency $\omega_c = 55 \pm 10$ GHz) with a coupling rate of $g = 7.5 \pm 1.2$ GHz. Three important parameters can be used to evaluate whether the system is in the strong coupling regime[52–54]. First, the ratio between the coupling strength and dissipation rates $g/\Gamma$. The coupling strength is an order of magnitude larger than the dissipation rates which separates the coupling from a weak Purcell effect $g/\Gamma<1$. Second, the value of cooperativity $C$ or coherence measurement parameter $U = (Cg/\omega_c)^{1/2}$. We demonstrated values of $C$ = 225 and $U$ = 5.5 in acoustic coupling which ranks it among the top of various physical platforms[53,54]. Third, the value of $g/\omega_c$ which was used to differentiate the coupling regimes from strong coupling (<0.1), ultra-strong coupling (0.1–1), and deep strong coupling (>1). We obtained a value of $g/\omega_c$ = 0.14, where the coupling strength is large and comparable to the natural frequency of the non-interacting parts. The large $f \times Q$ product for the metallic nanoresonators also mean that this system is attractive for ground state cooling from room temperature. Furthermore, all optical excitation and detection of mechanical vibrations could provide a way to dynamically manipulate phonon motion[55]. The plasmonic nanoresonators described above thus provide a platform for exploring novel phenomena, such as coupling-induced transparency in a purely mechanical system[21,22]. We believe that the metallic resonator system explored in this study is important not just for providing another physical platform to observe the strong coupling, but also providing an interdisciplinary study between plasmonics and optomechanics.

In summary, we have demonstrated strong vibrational coupling in plasmonic resonators. Engineering the phonon dissipation pathways by blocking the out-propagating acoustic waves improved the vibration quality factor an order of magnitude to $Q$ > 200 in Au nanoplates. We experimentally realized the highest frequency quality factor product $f \times Q = 1 \times 10^{13}$ Hz to date for plasmonic nanoresonators. The high-quality factors for these nanoresonators allowed us to observe strong vibrational coupling between different nanoplates. Analysis of the data using a coupled harmonic oscillator model gave an average coupling rate $g = 7.5 \pm 1.2$ GHz and cooperativity $C$ = 225 for the system. The metallic nanoresonators described in this study provide a platform for observation and control of quantum phonon dynamics.

## Methods

**Materials**. HAuCl$_4$·3H$_2$O, 1-pentanol, and PVP (Mn = 40,000) were purchased from Sigma-Aldrich (USA). Ethanol (AR, ≥99.7%) was purchased from Sinopharm Chemical Reagent Co., Ltd (Shanghai, China). Ultrapure water (18.2 MΩ × cm) was used throughout the experiments. Glass coverslips (catalog no. CG15KH) were purchased from Thorlabs China. Lacey carbon film with average pore size of ~5 μm and thickness of ~20 nm coated copper grids (catalog no. BZ110125b) were purchased from Electron Microscopy Supplies China.

**Au nanoplate synthesis**. The synthesis procedure was modified from previous studies[27]. Briefly, all glassware was cleaned with aqua regia and rinsed with deionized water before use. PVP (Mn = 40,000, 5 g) was dissolved into a mixture of 20 mL ultrapure water and 200 mL 1-pentanol and heated to 60 °C until fully transparent. Then, 50 μL HAuCl$_4$·3H$_2$O (0.2 M) ethanol solution, and 20 mL of 1-pentanol were sequentially added to 5 mL of the as-prepared mixture solution

while stirring. The solution was then heated to 120 °C and kept for 1 h under continuous stirring and another 3 h without stirring disturbance to facilitate the growth of Au nanoplates. The solution was brought to room temperature and the product was collected and washed with ethanol at least three times by centrifugation and ultrasonication to remove PVP surfactant. The Au nanoplates were ready for experimental measurements.

**Femtosecond time-resolved pump–probe spectroscopy.** Acoustic vibrations of the Au nanoplates were excited with femtosecond pulse lasers at 800 nm and detected at 530 nm. The experimental setup has been detailed elsewhere[29]. Briefly, the measurements were based on a Coherent Mira 900 Ti:sapphire oscillator laser system which gives output power of ~3.8 W at 800 nm with repetition rate of ~76 MHz and ~100 fs pulse width. The output laser beam was split into two portions with a 80/20 beamsplitter. The stronger portion of the beam was fed into an optical parametric oscillator (Coherent Mira OPO) to generate the probe light. The weaker portion was used to excite the Au nanoplates and modulated at 1 MHz by an acousto-optic modulator (IntraAction AOM-402AF3), triggered by the internal function generator of a lock-in amplifier (Stanford Research Systems SR844). The pump and probe beams were spatially overlapped with a dichroic beamsplitter and focused at the sample with an Olympus 60×, 0.9 numerical aperture (NA) microscope objective. Note that the two beams were both expanded before the lens to realize the full NA. The polarizations of the pump and probe beams were made linear and circular, respectively. In the current studies, measurements were all performed in reflection mode, with an avalanche photodiode (APD, Hamamatsu C12702-11) to detect the reflected probe beam. Transient reflectivity traces were recorded by monitoring the signal from the APD with the lock-in amplifier, with a time constant of 30 ms. A Thorlabs DDS600 linear translation stage was used to control the time delay between the pump and probe beams. The intensities of the pump and probe beams were controlled by half-wave plate and polarizer combinations. Typical powers were 3 mW for the pump and 100 μW for the probe. Under these conditions, the signal was stable and no melting or reshaping of the Au nanoplates was observed.

## Data availability
The source data underlying Fig. 1b, c, e, f, 2a, b, 3a, b and 4a, b, are provided as a Source Data file. Supplementary Figures and other images of this study are available from the corresponding authors upon reasonable request.

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

## Acknowledgements

This work was supported by the National Natural Science Foundation of China (NSFC) (Grant Nos. 61705133, 11734012, 11574218, 51502280), the Science and Technology Innovation Commission of Shenzhen (Grant No. JCYJ20170818143739628), and Natural Science Foundation of SZU (Grant No. 827/000267). G.V.H. acknowledges the support of the US National Science Foundation through award CHE-1502848. J.E.S. acknowledges support from the Australian Research Council Centre of Excellence in Exciton Science (CE170100026) and the Australian Research Council Grants Scheme.

## Author contributions

K.Y. and G.P.W. conceived the experiments. J.W. and K.Y. carried out the optical measurements. Y.Y. synthesized the Au nanoplates. K.Y. performed data analysis. J.E.S. and G.V.H. contributed theory. K.Y. wrote the paper with contributions from all the authors. G.P.W. supervised the entire project.

## Additional information

**Competing interests:** The authors declare no competing interests.

