## [Peer Review File · Nature Communications]

This manuscript has been previously reviewed at another journal that is not operating a transparent peer review scheme. This document only contains reviewer comments and rebuttal letters for versions considered at Nature Communications. Mentions of the other journal have been redacted.

Reviewers' Comments:

Reviewer #1:

Remarks to the Author:

The authors have considerably improved the manuscript. The definition of the coupling is now given (and it is correctly defined), so one can meaningfully compare the coupling to the damping rates. The authors present additional experimental data measured at different positions on the nanoplates, which allows them to do a statistical analysis and assign error bars to the observed quantities. These improvements make the reported observation of strong coupling convincing. The authors have also satisfactorily responded to my other concerns, and done further improvements based on the comments of other referees.

I have already stated what I think is novel and important in the manuscript in my previous report. I would like to add that it is valuable for science in general that this study combines in a novel way plasmonics and optomechanics, and introduces a system where time-domain studies of strong coupling are highly feasible. The manuscript is now, after revision, of high quality and presents novel findings of interest to a broad audience.

For all these reasons, I recommend publication in Nature Communications. However, I recommend that the authors consider first the following point: Several researchers indeed give $g/\omega > 0.1$ as a limit for ultrastrong coupling, but others stick to $g/\omega \sim 1$. I find the latter the more correct one, since the effects typical for the ultrastrong coupling regime become prominent only when $g/\omega \sim 1$. I find the authors claims about ultrastrong coupling kind of overselling, and would recommend the authors take away those claims (they can just say that the $g/\omega = 0.14$ is quite large). This would by no means decrease the quality of the manuscript, on the contrary.

Reviewer #2:

Remarks to the Author:

The authors have suitably addressed my concerns.

Reviewer #3:

Remarks to the Author:

The authors have addressed my technical concerns and part of my general concerns about the interest of the manuscript to a broad audience.

I find that the manuscript might be ready for publication in Nature Communications.

reviewers' comments

Reviewer #1 (Remarks to the Author):

The authors have considerably improved the manuscript. The definition of the coupling is now given (and it is correctly defined), so one can meaningfully compare the coupling to the damping rates. The authors present additional experimental data measured at different positions on the nanoplates, which allows them to do a statistical analysis and assign error bars to the observed quantities. These improvements make the reported observation of strong coupling convincing. The authors have also satisfactorily responded to my other concerns, and done further improvements based on the comments of other referees.

I have already stated what I think is novel and important in the manuscript in my previous report. I would like to add that it is valuable for science in general that this study combines in a novel way plasmonics and optomechanics, and introduces a system where time-domain studies of strong coupling are highly feasible. The manuscript is now, after revision, of high quality and presents novel findings of interest to a broad audience.

For all these reasons, I recommend publication in Nature Communications. However, I recommend that the authors consider first the following point: Several researchers indeed give $g/\omega > 0.1$ as a limit for ultrastrong coupling, but others stick to $g/\omega \sim 1$. I find the latter the more correct one, since the effects typical for the ultrastrong coupling regime become prominent only when $g/\omega \sim 1$. I find the authors claims about ultrastrong coupling kind of overselling, and would recommend the authors take away those claims (they can just say that the $g/\omega = 0.14$ is quite large). This would by no means decrease the quality of the manuscript, on the contrary.

Reply: We thank reviewer #1 for the constructive comments during the whole reviewing process.

In the final version, we have deleted the claims of ultrastrong coupling in our plasmonic system by just saying :“ We obtained a value of $g/\omega_c = 0.14$, where the coupling strength is large and comparable to the natural frequency of the non-interacting parts.”

Reviewer #2 (Remarks to the Author):

The authors have suitably addressed my concerns.

Reviewer #3 (Remarks to the Author):

The authors have addressed my technical concerns and part of my general concerns about the interest of the manuscript to a broad audience.

I find that the manuscript might be ready for publication in Nature Communications.

Reply: We thank both reviewers for their efforts and valuable comments of our manuscript.